# Machine Learning Approaches for Skin Cancer Classification from Dermoscopic Images: A Systematic Review

**Flavia Grignaffini** [1], **Francesco Barbuto** [1], **Lorenzo Piazzo** [1], **Maurizio Troiano** [1], **Patrizio Simeoni** [2], **Fabio Mangini** [1], **Giovanni Pellacani** [3], **Carmen Cantisani** [3] and **Fabrizio Frezza** [1,*]

1  Department of Information Engineering, Electronics and Telecommunications (DIET), "La Sapienza" University of Rome, 00184 Rome, Italy
2  National Transport Authority (NTA), D02WT20 Dublin, Ireland
3  Dermatology Unit, Department of Clinical Internal Anesthesiologic Cardiovascular Sciences, "La Sapienza" University of Rome, 00184 Rome, Italy
*  Correspondence: fabrizio.frezza@uniroma1.it

**Abstract:** Skin cancer (SC) is one of the most prevalent cancers worldwide. Clinical evaluation of skin lesions is necessary to assess the characteristics of the disease; however, it is limited by long timelines and variety in interpretation. As early and accurate diagnosis of SC is crucial to increase patient survival rates, machine-learning (ML) and deep-learning (DL) approaches have been developed to overcome these issues and support dermatologists. We present a systematic literature review of recent research on the use of machine learning to classify skin lesions with the aim of providing a solid starting point for researchers beginning to work in this area. A search was conducted in several electronic databases by applying inclusion/exclusion filters and for this review, only those documents that clearly and completely described the procedures performed and reported the results obtained were selected. Sixty-eight articles were selected, of which the majority use DL approaches, in particular convolutional neural networks (CNN), while a smaller portion rely on ML techniques or hybrid ML/DL approaches for skin cancer detection and classification. Many ML and DL methods show high performance as classifiers of skin lesions. The promising results obtained to date bode well for the not-too-distant inclusion of these techniques in clinical practice.

**Keywords:** skin cancer; skin lesion classification; melanoma classification; computer-aided diagnostics; artificial intelligence; machine learning; deep learning; convolutional neural networks

## 1. Introduction

Skin cancer is among the most common types of cancer in the Caucasian population worldwide [1]. It is one of the three most dangerous and fastest-growing types of cancer and therefore represents a significant public health problem [2]. According to the World Health Organisation, one out of every three cancer diagnoses is related to skin cancer [3] and according to the Skin Cancer Foundation, the global incidence of skin cancer continues to increase [4]. Skin tumours can be either benign or malignant; both types originate from DNA [5] damage due to ultraviolet radiation exposure that causes uncontrolled cell proliferation. Benign tumours, although they grow, do not spread. These include seborrhoeic keratosis, cherry angiomas, dermatofibroma, skin tags, pyrogenic granuloma, and cysts [6]. In contrast, malignant tumours expand in the patient's body, spread uncontrollably, and can infiltrate other tissues/organs. Below are the most frequent forms of cutaneous malignant tumours [7,8].

**Basal cell carcinoma or basalioma (BCC)** (Figure 1a). It accounts for about 80% of cases and originates in the basal cells, the deepest cells of the epidermis. Basal cell growth is slow, so in most cases BCC is curable and causes minimal damage if diagnosed and treated in time.

**Squamous cell carcinoma or cutaneous spinocellular carcinoma (SCC)** (Figure 1b). This accounts for approximately 16% of skin cancers and originates in the squamous cells in the most superficial layer of the epidermis. If detected early it is easily curable, but if neglected it can infiltrate the deeper layers of the skin and spread to other parts of the body.

**Malignant Melanoma (MM)** (Figure 1c). Originating in the melanocytic cells located in the epidermis, it is the most aggressive malignant skin tumour. It spreads rapidly, has a high mortality rate as it metastasises in the early stages, and is difficult to treat. It accounts for only 4% of skin cancers but induces mortality in 80% of cases. Only 14% of patients with metastatic melanoma survive for five years [9]. If diagnosed in the early stages it has a 95% curability rate, so its early diagnosis can greatly increase life chances.

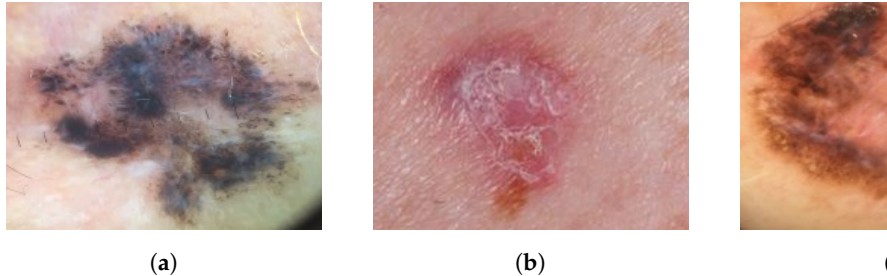

| (a) | (b) | (c) |

**Figure 1.** Principal types of malignant skin cancer (sources: [10] and Dermatology Unit, Department of Clinical Internal Anesthesiologic Cardiovascular Sciences, "La Sapienza" University of Rome). (**a**) BCC. (**b**) SCC. (**c**) MM.

Although it accounts for the minority of skin cancer cases, melanoma is the most aggressive form of skin cancer with an increasing incidence rate. It can prove lethal if not diagnosed in time, so it is crucial to detect it early in the process to increase the chance of cure and recovery [11]. The rules currently used by dermatologists to diagnose melanoma are summarized in Table 1.

**Table 1.** Summary of the melanoma diagnosis rules.

| Diagnosis Rules | Description |
| --- | --- |
| **The ABCDE rule [12,13]** | It is based on morphological characteristics such as asymmetry (A), irregularity of the edges (B), nonhomogeneous color (C), a diameter size (D) greater than or equal to 6 mm, and evolution (E) understood as temporal changes in size, shape, color, elevation, and the appearance of new symptoms (bleeding, itching, scab formation) [14]. |
| **Seven Point Checklist [15]** | It is based on the seven main dermoscopic features of melanoma (major criteria: atypical pigment network, blue-whitish veil and atypical vascular pattern; minor criteria: irregular pigmentation, irregular streaks, irregular dots and globules, regression structures) by assigning a score to each of these. |
| **The Menzies method [16]** | It is based on 11 features, two negative and nine positive, which are assessed as present/absent. |

One of the main tools for the early diagnosis of melanoma is dermoscopy, a noninvasive and cost-effective technique [17,18], that has proved useful in reducing the number of presumptive diagnoses that need to be confirmed histologically by skin biopsy [19]. The equipment magnifies up to 10 times over the area of interest and this allows the physician to obtain what would be possible by employing only the naked eye [20], thus facilitating the detection of certain features of the lesions that are essential for diagnosis, such as symmetry, size, broths, and presence and distribution of color features, but also blue–white areas, atypical pigmented networks and globules [21]. It is, however, a complex, time-consuming procedure that shows a strong dependence on the experience and subjectivity of the physician. The issues illustrated above made necessary the development of computer-aided

diagnostic systems (CAD systems). Those systems involve the steps shown in Figure 2 for analysing and classifying dermoscopic images of skin lesions [22].

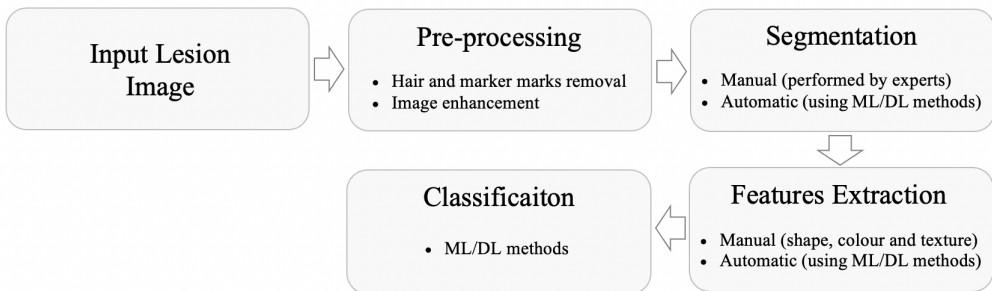

**Figure 2.** CAD's pipeline for skin lesion image analysis.

Preprocessing is aimed at mitigating artefacts in the images, mainly due to the presence of hair and marker marks on lesions. A typical hair-removal algorithm comprises two steps [23]: hair detection, for which various morphological, thresholding and filtering operations (Gaussian, middle, and median filters) are applied, and hair repair (restoration or "inpainting"). The latter, which consists of filling the image space occupied by removed hair, is performed by means of linear interpolation techniques, nonlinear partial differential equations (PDEs), diffusion methods, or exemplar-based methods. There are also well-known hair-removal algorithms, such as DullRazor [24]. Among image-enhancement methods, the most important are color correction or calibration, which recover real colors of a lesion, but there are also illumination-correction, contrast-enhancement, and edge-enhancement techniques. Illumination correction is performed by illumination-reflectance models, Retinex algorithm [25], and bilateral filter or Monte Carlo sampling [26]. For contrast enhancement, the equalization histogram (HE), the equalization of the adaptive histogram (AHE) and a sharp masking are often used together [25]. Finally, for edge enhancement, the Karhunen–Loève Transform (KLT), also known as the hoteling transform or principal component analysis (PCA), is widely used [27]. The segmentation step is crucial to increase the effectiveness of subsequent steps as clinically important features, such as blue–white areas, atypical pigmented networks and globules, can only be automatically extracted when the accuracy of lesion edge detection is high [28]. This is a crucial task that researchers need to perform to aim for the best results [7,29–38]. The feature-extraction step can be either manual [39] or automated by means of machine-learning algorithms. The extraction of handcrafted features relevant in the case of skin lesion classification is based on the methodologies designed by dermatologists to perform skin cancer diagnosis, and, in particular, the ABCD rule of dermoscopy. The main operations used for the extraction of shape, color, and texture features of skin lesions are given below.

- Shape: computation of area, perimeter, compactness index, rectangularity, bulkiness, major and minor axis length, convex hull, comparison with a circle, eccentricity, Hu's moment invariants, wavelet invariant moments, Zunic compactness, symmetry maps, symmetry distance, and adaptive fuzzy symmetry distance.
- Color: computation of average, standard deviation, variance, skewness, maximum, minimum, entropy, 1D or 3D color histograms, and the autocorrelogram. In addition, several techniques have been used to group the pixels, namely k-means, Gaussian mixture model (GMM), and multi-thresholding.
- Texture: computation of the gray-level co-occurrence matrix (GCLM), gray level run-length matrix (GLRLM), local binary patterns (LBP), wavelet and Fourier transforms, fractal dimension, multidimensional receptive fields histograms, Markov random fields, and Gabor filters.

By using machine-learning methods, learned features are derived automatically from the datasets and require no prior knowledge of the problem. Even for the final classification phase, different approaches are possible, from the classical ones, to the cutting-edge

methodologies based on deep convolutional neural networks. The techniques used to classify skin lesions are similar to those used for other types of cancer, such as breast, thyroid, colorectal, lung, pancreatic, and cervical cancers [40–44]. In this review article, various approaches are examined to then determine which show the best performance in the tasks of skin lesion classification and skin cancer detection.

Many studies show that the performance of DL algorithms equals or even exceeds the performance of experienced dermatologists in detecting and diagnosing skin lesions [45–52]. However, the performance of these algorithms should also be evaluated on images outside their area of expertise [45]. Several difficulties and challenges exist in the automatic classification of dermoscopic images using ML and DL methods [53], such as high variability in the shape, size, and location of lesions, the low contrast between skin lesions and surrounding healthy skin, the visual similarity between melanoma and nonmelanoma lesions, and the variation in skin condition among different patients. Regarding this last point, a very important but little addressed aspect is skin color [54]. In fact, the dermoscopic datasets used for training machine-learning models contain images of light-skinned people. To perform accurate detection of skin lesions in dark-skinned people, it is necessary to expand the existing datasets and fill this gap. Opportunities of the use of ML and DL methods for skin cancer detection, in addition to those already mentioned, include the possibility of avoiding unnecessary biopsies or missed melanomas, but also of making skin diagnoses without the need for physical contact with the patient's skin and reducing the cost of diagnosis and treatment of nonmelanoma skin cancer, which is found to be considerable [55].

The paper is organised into the following sections.

- Section 2. We present the methodology employed to perform the systematic research and present the main public databases containing dermoscopic images, relevant for the paper analysed here.
- Section 3. In this section, we discuss and explain several ML and DL methods commonly used for demoscopic image classification tasks.
- Section 4. We summarise in this section all the research applied to skin lesions on dermoscopic images selected for this paper; those works are categorised according to the approach taken, i.e., ML, DL, and ML/DL hybrid.
- Section 5. In this section, results are discussed.

## 2. Material and Methods

### 2.1. Search Strategy

This systematic review presents the work conducted over the last decade on skin cancer classification using ML and DL techniques with the aim of providing an overview of the problem and possible solutions to those who wish to approach this very important and extremely topical issue. For the article selection phase, the following keywords were inserted in the search field of the electronic databases arXiv and ScienceDirect to be combined with the logical operators "and" and "or": melanoma, detection, classification, machine learning, deep learning, dermoscopic images. Study inclusion and data extraction are in accordance with the preferred reporting items for systematic reviews and meta-analyses (PRISMA) guidelines (Figure 3) [56].

In the selection, inclusion criteria were applied such as (i) openly published articles, (ii) publications in English, (iii) classification papers, (iv) papers based on dermoscopic images and (v) articles published between 2012 and 2022. Exclusion criteria were also applied: (i) review articles, (ii) articles published in a language other than English, (iii) articles not complete with results, (iv) articles dealing only with segmentation, and (v) articles not using public datasets. Using these criteria, 68 research articles were collected.

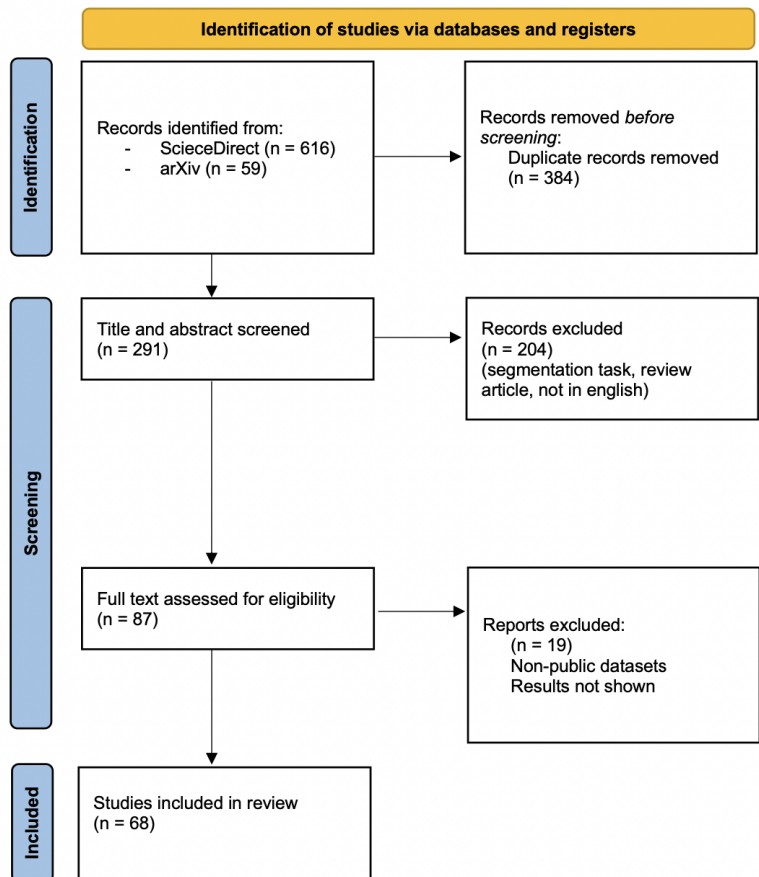

**Figure 3.** PRISMA flow diagram.

*2.2. Common Skin Lesion Databases*

With the aim of implementing CAD systems in dermatology and testing them on consistent real data, several dermoscopic image datasets were collected. The most common dermoscopic public datasets are introduced below, and their details are summarised in Table 2 [8].

**ISIC archive**. The ISIC archive [10], which combines several datasets of skin lesions, was originally released by the International Skin Imaging Collaboration in 2016 for the challenge called International Symposium on Biomedical Imaging (ISBI). Various modifications have been made over the years.

Kaggle, one of the best resources for data scientists and machine learners looking for datasets, collected several databases based on the ISIC archive.

**HAM10000**. The human-against-machine dataset (HAM) [57] (available at [58]), that arises from the addition of some images to the ISIC2018 dataset, contains more than 10,000 images with seven different diagnoses collected from two sources: Cliff Rosendahl's skin cancer practice in Queensland, Australia, and the Dermatology Department of the Medical University of Vienna, Austria.

**PH²**. The PH² database [59] (available at [60]) acquired at the Dermatology Service of Hospital Pedro Hispano, Matosinhos, Portugal, contains 200 images divided into common nevi, atypical nevi, and melanoma skin cancer images. Together with the images, annotations such as medical segmentation of the pigmented skin lesion, histological and clinical diagnoses, and scores assigned by other dermatological criteria are provided.

**MedNode**. The MedNode dataset [61] (available at [62]) contains images of skin lesions in the category of melanoma and common nevus from the digital image archive of the Department of Dermatology of the University Medical Center Groningen (UMCG).

**Table 2.** Summary of the most common public skin lesion datasets, which contain images of nevi (N)/atypical nevi (AN), common nevi (CN), malignant melanomas (MM), seborrheic keratoses (SK), basal cell carcinomas (BCC), dermatofibromas (DF), actinic keratoses (AK), vascular lesions (VL), and squamous cell carcinomas (SCC).

| Database | N/AN | CN | MM | SK | BCC | DF | AK | VL | SCC | Tot |
|---|---|---|---|---|---|---|---|---|---|---|
| ISIC 2016 [63] | 726 | - | 173 | - | - | - | - | - | - | 899 |
| ISIC 2017 [64] | 1372 | - | 374 | 254 | - | - | - | - | - | 2000 |
| ISIC 2019 [65] | 12,875 | - | 4522 | 2624 | 3323 | 239 | 867 | 253 | 628 | 25,331 |
| ISIC 2020 [66] | 27,124 | 5193 | 584 | 135 | - | - | - | - | - | 33,126 |
| HAM10000 | 6705 | - | 1113 | 1099 | 514 | 115 | 327 | 142 | - | 10,015 |
| PH$^2$ | 80 | 80 | 40 | - | - | - | - | - | - | 200 |
| MedNode | 100 | - | 70 | - | - | - | - | - | - | 170 |

As introduced earlier, the inclusion criteria for article selection include the use of public dermoscopic datasets.

## 3. Artificial Intelligence

Artificial intelligence (AI) refers to the ability of machines to perform some of the tasks characteristic of human intelligence including planning, problem solving, natural language understanding, and learning. The two main branches of AI are discussed in the following paragraphs. In addition, the main pre-trained networks are presented and the concept of transfer learning (TL) is introduced.

### 3.1. Machine Learning

Machine learning is the application and science of algorithms that autonomously extract useful information from data. In the training process, the ML model receives the training data as input and processes it to extract natural patterns and/or salient features based on which it learns to associate an attribute with each sample or assign each sample to one of the identified clusters. This will allow the model to make predictions on new data never seen before. The main ML models are outlined below.

#### 3.1.1. Decision Trees

Decision Trees (DT) are versatile ML algorithms that work for both categorical and numerical variables since they do not require an assumption about the data distribution and classifier structure. Thus, these algorithms can perform classification, regression and multi-output tasks. They provide accurate and efficient classifications for large and complex datasets. Random forests (RF) are based on the ensemble of decision trees. By using multiple DTs, which individually suffer from high variance, RFs constitute a more robust model that offers better generalization performance [67].

#### 3.1.2. Support Vector Machines

Support vector machines (SVMs) are ML models capable of performing linear and nonlinear classifications (kernel methods), regressions and outlier detection [68,69]. Although the datasets on which SVMs perform well can be complex, and these need to be not too large. When applied to classification tasks, SVMs build hyperplanes. Every hyperplane represents a decision boundary that allows the separation and differentiation of the feature space in two distinct classes. When the data are linearly separable, linear classification can be performed, whereas if the data are not linearly separable, a kernel function (linear, quadratic, cubic, fine Gaussian, medium Gaussian, coarse, etc.) can be selected to map the data into a higher dimensional space with the goal of forcing the data points to become linearly separable, if possible.

#### 3.1.3. K-Nearest Neighbors

The K-Nearest Neighbour (KNN) is an algorithm that performs classification of new data based on its similarity to the closest labelled data [70,71]. Once the parameters

associated with a KNN classifier have been chosen, i.e., the K-number of nearest neighbours to be considered and the distance metric (of which the most significant are the Euclidean and Manhattan Distances), the new data is assigned a label based on a majority vote. To overcome overfitting and underfitting problems, a value of K between 3 and 10 is typically chosen.

### 3.1.4. Artificial Neural Networks

Artificial neural networks (ANNs), developed from studies related to neuronal connections, were introduced to solve problems as complex as real ones. Early neural networks attempted to mimic the human brain and the synaptic connections between neurons, but being able to understand only part of the brain mechanisms, ANNs were implemented by means of simpler and more ordered architectures composed of functional units called neurons (or nodes), connected by arcs simulating synaptic connections, and structured in layers. The layers, which are the main element of neural networks, extract representations from the data, meaningful to the problem at hand, and process them. What gives a neural network its learning capability is the ability to adjust the weights associated with the connections between neurons during training (i.e., based on the experience gained). The methodology used in training neural networks is called the learning paradigm.

### 3.2. Deep Learning

Deep learning (DL), the most successful ML solution, is an intelligent algorithm capable of autonomously learning from a dataset by exploiting a complex architecture that simulates the human brain structure. Since the early 2000s, convolutional neural networks (CNNs or ConvNet), inspired by the biological neural networks of the visual cortex [72], have become the most effective and widely used algorithms in computer vision. The layers of CNNs are of different types and each has its own specific function. In fact, some of them have trainable parameters, while others only have the task of implementing an established function. The types of layers most frequently used in CNN architectures [73] are as follows.

**Convolutional layers.** Convolutional layers are able to learn local patterns, and this entails two important properties: the learned patterns are translation invariant and the learning extends to spatial hierarchies of patterns. This allows the CNN to efficiently learn increasingly complex visual concepts as the depth of the network increases. Convolutional layers contain a series of filters that run over the input image performing the convolution operation and generate feature maps to be sent to subsequent layers.

**Normalization layers.** These are layers for normalising input data by means of a specific function that does not provide any trainable parameters and only acts in forward propagation. The use of those layers has diminished in recent times.

**Regularization layer.** They are layers designed to reduce overfitting by randomly ignoring a proportion of neurons during each training session. The best known regularization technique is the dropout.

**Pooling layers.** Pooling layers perform subsampling of feature maps while retaining the main information contained therein, in order to reduce the model parameters and the computational cost of the operations to be performed. Pooling filters, of which the most used are average pooling and max pooling, run over the feature maps they receive as input by performing the convolution operation as in the case of convolutional layers, but in this case there are no trainable parameters.

**Fully connected layers.** In those layers, every neuron of the layer is connected to all activation functions of the previous layer. The first fully connected layer (FC) takes input feature maps as output from the last convolutional or pooling layer, and the last of the FC layers is the CNN classifier.

### 3.3. Pre-Trained Models and Transfer Learning

Several CNN architectures are available as pretrained models. The most commonly used ones are GoogLeNet, InceptionV3, ResNet, SqueezeNet, DarkNet, DenseNet, Xception, Inception-ResNet, Nasnet, and EfficientNet, and the parameters of each are summarised in Table 3.

**Table 3.** Overview of common CNNs architectures.

| Architetture | Year | Developed by | Parameters | Layers | Input Size |
|---|---|---|---|---|---|
| GoogLeNet | 2014 | Szegedy et al. | 4 M | 144 | 224 × 224 |
| InceptionV3 | 2015 | Szegedy et al. | 23.8 M | 316 | 299 × 299 |
| ResNet18 | 2015 | He et al. | 11.17 M | 72 | 224 × 224 |
| ResNet50 | 2015 | He et al. | 25.6 M | 177 | 224 × 224 |
| ResNet101 | 2015 | He et al. | 44.7 M | 347 | 224 × 224 |
| SqueezeNet | 2016 | Iandola et al. | 1.2 M | 68 | 227 × 227 |
| DenseNet201 | 2017 | Huang et al. | 20.2 M | 709 | 224 × 224 |
| Xception | 2017 | Chollet | 22.9 M | 171 | 299 × 299 |
| Inception-ResNet | 2017 | Szegedy et al. | 55.8 M | 824 | 299 × 299 |
| EfficientNetB0 | 2019 | Mingxing and Le | 5.3 M | 290 | 224 × 224 |

As introduced earlier, training DL models with randomly initialised parameters requires a large amount of labelled data, which is often not readily available. Transfer learning represents the best solution by allowing the reuse of knowledge (the weights) extracted from a pretrained CNN model on large datasets labelled as ImageNet and achieving good results in the source domain. Transfer learning can be used as a feature extraction model but also to refine hyperparameters by freezing or unfreezing various layers.

## 4. Results

This section summarises the papers on skin lesion classification collected from the literature after careful research. Starting with papers that propose only ML techniques for skin cancer detection, we continue with papers that focus on DL techniques, and finally we report those that combine ML and DL. The following metrics were used to evaluate model performance: accuracy (ACC), sensitivity (SE), specificity (SP), precision (PR), recall (REC), F1 score (F1), and area under the ROC curve (AUC).

### 4.1. Machine-Learning Methods

A new algorithm for calculating the extended feature vector space is proposed in [74]. Specifically, features of color (from the hue-saturation value (HSV) space) and texture (via local binary pattern (LBP)) are extracted from the images and subsequently combined to extend the feature space. These features are then used by an ensemble bagged tree classifier for the detection of melanoma.

In one paper [75], the authors, after smoothing the images with a Gaussian filter, use the active contour model to obtain the lesion edges from which they define a segmentation mask to extract lesion characteristics in terms of shape. From the mask, they replace the lesion pixels with those of the original image and then extract lesion characteristics in terms of color and texture. Finally, they use a K-nearest neighbor (KNN) model to perform the binary classification between the melanoma class and the seborrhoeic nevi-keratosis class. They obtain the best results with $k = 2$.

In [76], lesion segmentation is performed in the gray space whereas in the RGB color space, texture characteristics are extracted with global (grey-level co-occurence matrix (GLCM) for entropy, contrast, correlation, angular second moment, inverse different moment, sum of squares) and local (LBP and oriented FAST and rotated BRIEF (ORB)) techniques. To extract color features from each image, histograms of the five color spaces (grayscale, RGB, YCrCb, L*a*b and HSV) are generated from which information on mean, standard deviation, skewness and kurtosis is obtained for a total of 52 color features for each image. In order to select only the most significant features, two variants of the Harris Hawk optimisation (HHO) algorithm are tested, employing S-shaped (BHHO-S) and V-

shaped (BHHO-V) transfer functions respectively, with BHHO-S leading to better results. As a final step, an SVM model is used to perform melanoma/non-melanoma classification on dermatological images.

The authors of [77] use the lesion masks already provided with the images to extract 510 features (18 for shape, 72 for color and 420 for texture) which are then manipulated to create different subsets of features and sent to ensemble classification models that will use them to diagnose skin lesions. Each ensemble classification model is generated by using an optimum-path forest (OPF) classifier and integrated with a majority voting strategy. Three different approaches are proposed: SE-OPS, which manipulates features by using different subsets based on specific feature groups, SEFS-OPF, which manipulates features by using correlation-based features selection (CFS) to select the best features, and FEFS-OPS, which manipulates features by using different selection algorithms such as correlation coefficient information gain, principal component analysis (PCA) and CFS. The best model is found to be SE-OPF.

In [78], shape (normalised radial length (4 features), asymmetry of shape (2 features)), color (statistical color measures (12 features), six-color model (7 features)), and texture (statistical texture measures (8 features), energy of Laws' filters responses (14 features), gray-level co-occurrence matrix features (24 features)) are extracted from all areas of the lesion (general features), as well as some texture features from peripheral regions only (local features). The sequential feature selection (SFS) approach is used and then classification is performed by using two different models: on the one hand a linear SVM model to recognise melanoma versus nevus on the basis of four lesion features, and on the other hand the RUSBoost classifier to recognise melanoma versus nevus and atypical nevus on the basis of eight features considered relevant by the SFS algorithm.

In [79], an original and innovative system for automatic melanoma skin detection (ASMD) with melanoma index (MI) is proposed. The system incorporates image pre-processing, bi-dimensional empirical mode decomposition (BEMD), image texture enhancement, entropy and energy feature extraction and binary classification. From the feature-extraction stage, vectors of 28 features are obtained for each image, and a Student's *t*-test with triple cross-validation is used to classify the 28 features based on the statistical results obtained. In the classification phase, the combination SVM and radial basis function (RBF) offers high accuracy, which prompts the authors of the paper to formulate a clinically relevant MI based on Rényi entropy and maximum entropy. The MI value can help dermatologists decide whether a suspected skin lesion, shown in dermoscopic images, is benign or malignant.

In [80], an integrated computer-aided method for multiclass classification of melanoma, dysplastic nevi, and basal cell carcinoma is proposed. Different features related to shape, edge irregularity, color and texture (obtained by combining GLCM and a fractal-based regional texture analysis (FRTA)) of skin lesions are extracted. Finally, the combination of feature selection with recursive feature elimination method (RFE) and a SVM with RBF function is used to perform classification.

The paper [81] addresses the problem of amorphous pigmentary lesions and blurred edges by proposing two new fractal signatures called $S_{STF}$ statistical fractal signatures and $S_{SPF}$ statistical prism-based fractal signatures. The comparison of different computer-aided diagnosis methods for multiclass skin lesion classification based on the new fractal signatures, and using different classifiers, is performed. The best results for robust, unbiased, and reproducible methodologies are obtained by using $S_{STF}$ with the LDA classifier.

In [82], the authors suggest a skin lesion segmentation and classification system based on sparse kernel representation. They use a kernel dictionary and classifier to predict the labels of the test set. In particular, they first extract the texture features (speeded up robust features, or SURF) from the images, then, by using the KOMP algorithm, compute the sparse code of it with respect to the kernel dictionary, and finally, the classifier is used to predict the class of the lesion. They perform both binary classification (melanoma/normal) and multiclass classification (melanoma, basal cell carcinoma, and nevi).

The authors of [83] propose a methodology for the accurate diagnosis of melanoma from dermoscopic images that consists of extracting and selecting salient features from the preprocessed and segmented images and classifying them by using multilayer perceptron (MPL)-averaged. Both the feature extraction and classification steps are optimized by a newly developed version of the red fox optimization (DRFO) algorithm.

In this work [84], the authors perform skin lesion segmentation by using a novel dynamic graph cut algorithm, extract texture (contrast, correlation, energy, homogeneity, and entropy), color (mean, standard deviation, skeweness, and variance), and asymmetry (asymmetry and bulkiness) features from a segmented skin region, and then use a probabilistic classifier called Naïve Bayes for skin disease classification.

Table 4 summarizes the machine-learning methods previously described.

**Table 4.** Overview of cited works using ML approaches (results have been rounded). $F_{EX}$ and $F_{SE}$ abbreviations are used for feature extraction and selection, respectively. The symbol "-" indicates that no information is provided on a particular operation.

| Author & Year | Classification Task | Dataset | Data Augmentation | Methods Used | Cross Validation | Results |
|---|---|---|---|---|---|---|
| **Kumar et al. [74]** 2022 | Binary: MM vs. benign | MedNode | - | $F_{EX}$ + Ensemble Bagged Tree classifier | - | ACC = 0.95, SE = 0.94, SP = 0.97, AUC = 0.99 |
| **Kanca et al. [75]** 2022 | Binary: MM vs. N and SK | ISIC2017 | - | $F_{EX}$ + KNN classifier | - | ACC = 0.68, SE = 0.80, SP = 0.80 |
| **Bansal et al. [76]** 2022 | Binary: MM vs. non-MM | HAM10000 | Blurring, increased brightness, addition of contrast and noise, flipping, zoom, and others | $F_{EX}$ and $F_{SE}$ (with BHHO-S algorithm) + linear SVM | - | ACC = 0.88, SE = 0.89, SP = 0.89, PR = 0.86 |
| **Oliveira et al. [77]** 2017 | Binary: benign vs. malignant | ISIC2016 | - | $F_{EX}$ and $F_{SE}$ (using SE-OPS approach) + OPF classifier | 10-fold | ACC = 0.94, SE = 0.92, SP = 0.97 |
| **Tajeddin et al. [78]** 2018 | Binary: MM vs. N and MM vs. N/AN | PH$^2$ | - | $F_{EX}$ and $F_{SE}$ (with SFS approach) + linear SVM and RUSBoost classifiers | 10-fold | 1° SE = 0.97, SP = 1; 2° SE = 0.95, SP = 0.95 |
| **Cheong et al. [79]** 2021 | Binary: benign vs. malignant | DermIS, DermQuest, ISIC2016 | Image rotation: ±30, ±60 and ±90 degrees | $F_{EX}$ and $F_{SE}$ (t-Student test) + RBF-SVM | - | ACC = 0.98, SE = 0.97, SP = 0.98, PR = 0.98, F1 = 0.98 |
| **Chatterjee et al. [80]** 2019 | Multi-class: MM, N and BCC | ISIC archive, PH$^2$, IDS | - | $F_{EX}$ and $F_{SE}$ (RFE method) + RBF-SVM | 10-fold | ISIC: ACC = 0.99, SE = 0.98, SP = 0.98; PH$^2$: ACC = 0.98, SE = 0.91, SP = 0.99; IDS: ACC = 1, SE = 1, SP = 1 |
| **Camacho-Gutiérrez et al. [81]** 2022 | Multi-class: N, MM, SK, BCC, DF, AK, VL | ISIC 2019 | - | $S_{STF}$ statistical fractal signatures + LDA classifier | - | Four-classes: ACC = 0.87, SE = 0.63, SP = 0.89, PR = 0.65; seven-classes: ACC = 0.88, SE = 0.41, SP = 0.92, PR = 0.46 |
| **Moradi et al. [82]** 2019 | Binary: MM vs. normal; Multi-class: MM, BCC and N | ISIC2016, PH$^2$ | - | $F_{EX}$ and calculation of sparse code using KOMP algorithm + linear classifier | 10-fold | Binary ISIC: ACC = 0.96, SE = 0.97, SP = 0.93; binary PH2: ACC = 0.96, SE = 100, SP = 0.92; three-classes: overall ACC = 0.86 |
| **Fu et al. [83]** 2020 | Multi-class: BCC, SK, MM, N | ISIC2020 | - | $F_{EX}$ and $F_{SE}$ + MPL-averaged optimized by DRFO algorithm | - | ACC = 0.91, SE = 0.90, SP = 0.92 |
| **Balaji et al. [84]** 2020 | Multi-class: benign vs. malignant | ISIC2017 | - | $F_{EX}$ + Naïve Bayes classifier | - | ACC = 0.94 for benign cases, 0.91 for MM and 0.93 for SK. |



*4.2. Deep-Learning Methods*

In [85], a parameter transfer of a pretrained network to a CNN is performed to reduce the training time. The performance of the network without and with fine tuning (FT) is compared, obtaining better results in the second option.

For the melanoma detection task, an ensemble learning approach is proposed in [86] to combine the predictive power of three different deep convolutional neural network (DCNN) models known from medical imaging classifications pretrained on the ImageNet dataset: EfficientNetB8, SEResNeXt10, and DenseNet264. Two innovative approaches are used: the multisample dropout approach, whereby, downstream of the pre-trained network architectures, the dropout, fully connected (FC), and softmax layers are duplicated and the loss value (obtained by using a variant of the binary cross-entropy called focal loss to perform dense object detection) is calculated as the average of the loss values of all dropout samples, and, secondly, the multi-penalty approach, whereby each duplicated layer is penalised at a different rate.

Moreover, in [87], an ensemble learning approach is used. An ensemble of deep model (SLDEP) is created by using four different CNNs (GoogLeNet, VGGNet, ResNet, and ResNeXt) to perform multiclass classification based on majority voting.

In [88], the authors perform a multiclass classification by using TL on InceptionV3, ResNet50, and Denset201, removing the output layer from these architectures and adding pooling and FC layers.

In [89], the authors, after preprocessing the images to remove hair and improve image quality by using the HR-IQE (hair-removal image-quality enhancement) algorithm, proceed with lesion segmentation by using swarm intelligence (SW) algorithms to identify the region of interest (ROI), extract features within the ROI by using sped-up robust features (SURF) and select only a few of these based on the grasshopper optimisation algorithm (GOA). Finally, a custom CNN, consisting of two convolutional layers followed by two max-pooling layers, and a flatten layer, is used to classify images into melanoma and nonmelanoma classes.

In [90], an AWO-based SqueezeNet is proposed in which the pre-trained SqueezeNet is trained by a proposed AWO algorithm which is a fusion of the aquila optimisation (AO) algorithm and the whale optimisation algorithm (WOA).

In [91], a custom CNN with five convolutional layers, five max pooling layers, two dense layers and one dropout layer is used. The authors focus heavily on image preprocessing work to enable the network to achieve better performance.

In [92], various types of CNNs (ResNet, DenseNet, InceptionV3, VGG16) pretrained on ImageNet are implemented to evaluate their performance in the skin cancer diagnosis task. After selecting some features of the InceptionV3 and DenseNet architectures, a new architecture called DenseNet-II is built in which there are two parallel networks of convolutional layers. By using focal loss, they create an imbalance of weights to penalise the majority class and reduce the damaging effects of class imbalance.

In [93], a shallow DL model called $SCNN_{12}$ is created, consisting of 12 weighted layers: 4 convolutional, 4 max pooling, 1 flatten, 2 dense, and 1 softmax layer. The ablation study method is used to determine the parameters and hyperparameters of the model on the basis of optimal performance in terms of accuracy. In addition to classical preprocessing operations, the authors perform downsampling by reducing the spatial resolution of the images while keeping the size unchanged. In this way, the images retain their $224 \times 224$ size but are reduced from 45 kb to 6 kb spatial resolution.

An early skin cancer detection approach using a pretrained DL model is proposed in [94]. In this work, a Flask website is also developed to allow users to upload dermatological images and make a prediction on the class they belong to.

In [95], a new DL model is proposed based on the VGG16 architecture by eliminating some redundant convolutional layers, introducing a batch normalisation (BN) layer after each pooling layer and replacing the FC layer with a global average pooling (GAP) layer. Eliminating some convolutional layers decreases the trainable parameters and introducing

BN and GAP layers improves performance without increasing the number of parameters. By decreasing the network parameters compared to VGG16, the entire architecture is optimised and calculation times are accelerated.

With the idea of improving the existing performance measures and minimising the convergence time of the learning model in the skin cancer detection task, in [96] the authors use AlexNet as a pretrained architecture and replace its larger filters with smaller ones. This reduces the parametric complexity of the model but increases its depth, causing the vanishing gradient phenomenon during the training phase. To overcome this problem, residual or skip connections are introduced through several pairs of consecutive blocks (taking a cue from ResNet). Finally, learning rate annealing is applied by using the cyclic learning rate during training.

In [97], adversarial training is used to achieve good accuracy in skin tumour classification, despite having a small amount of data available. By applying the fast gradient sign method (FGSM), new adversarial example images are created to maximise the loss for the input image, which are subsequently used in both train and test phases. With these new images, some pretrained networks (VGG16, VGG19, DenseNet101, and ResNet101) are retrained, and ResNet101 obtains the best results even though it consumes more computational power and takes longer than the others.

In [98], the authors use the pretrained ResNet52 network in five different situations to classify skin lesions. The tests performed are: training without data augmentation (DA), training with DA only on malignant images, training with DA on malignant and downsampling (DS) of benign images with two different proportions, training with DA only on malignant images by including other images from different datasets in the dataset. The best solution appears to be the one in which the data is augmented only on lesions belonging to the malignant class while maintaining a malignant/benign ratio of 0.44.

In [99], the VGG16 network is used in three different ways: training from scratch, transfer learning, and fine tuning. The training-from-scratch approach turns out to be the least accurate of the three proposed. The TL method greatly outperforms the former but shows very different performance between the training and test phases, testifying to the presence of overfitting. Applying fine-tuning results in the best model with superior performance to the former approach and no evidence of overfitting on the train data.

A new approach that not only classifies skin lesions with DL models but also discriminates is proposed in [100]. An architecture is created to take a pair of images (malignant/benign) as input and use a light network pretrained on the ImageNet dataset to extract two feature vectors, one from each image, used individually to train two networks for melanoma recognition, and jointly to introduce a nonparametric discriminant layer through which a network is constructed to check whether or not the images corresponding to the two jobs belong to the same category.

In [101], after performing image preprocessing, segmentation and DA, they use the ResNet50 and InceptionV3 networks pretrained on ImageNet to perform binary classification of skin lesions. The final result is obtained by averaging the predictions generated by each classifier.

A deep clustering approach based on the incorporation in latent space of dermoscopic images of skin lesions is proposed in [102]. To learn discriminative embeddings, clustering is achieved by using a novel centre-oriented margin-free triplet loss (COM-Tripletenforced on image embedding from a CNN backbone). This variant of triplet loss is used because, in contrast to the classical one that maintains a fixed distance from the origin independently for positive and negative classes, it adaptively updates the distance between clusters during the training procedure. The method seeks to maximise the distance between cluster centres instead of minimising the classification error by making the model less sensitive to imbalance between classes. Furthermore, to get away from the need for labels, an unsupervised approach is proposed by implementing COM-Triplet loss on pseudo-labels generated by Gaussian mixture model (GMM). The CNN has an architecture based on the backbones common in computer vision tasks (VGG16, ResNet50, DenseNet169 and

EfficientNetB3) by replacing the dense layer with an embedding layer for deep clustering models. A dropout layer with a rate of 0.3 is also inserted between the backbone of the networks and this last layer. The best results are obtained by using the pretrained VGG16 network as the backbone and performing transfer learning.

To automatically detect skin cancer on dermoscopic images, in [103] the authors use a metalearning method (also known as "learning to learn") that aims at understanding the learning process in order to use the acquired knowledge to improve the learning effectiveness of new tasks. The authors demonstrate that nonmedical image features can be used to classify skin lesions and that the distribution of data affects the performance of the model. They use a pretrained ResNet50 by removing the last dense layer and perform cross-validation three times.

In [104], three pretrained networks (EfficientNet, SENet, and ResNet) are used in three different situations: training with preprocessed images, training with images multiplied by the segmentation mask obtained with the U-Net, and training with both of the previous solutions. The latter approach turns out to be the best in terms of accuracy.

In [105], the MobileNet network pretrained on images from the ImageNet dataset and optimised on dermatological images is used.

Several pre-trained neural networks (PNASNet-5-Large, InceptionResNetV2, SENet154 and InceptionV4) are being tested in [106], freezing all levels except the last FC where the softmax function is used to produce the probability of each class. The best result in terms of accuracy is achieved by the model based on the PNASNet-5-Large network.

In [107], transfer learning (TL) is performed on the VGG16 and GoogLeNet networks, which are evaluated both individually and combined. The best result is obtained with the combination of the two models.

A multitask deep learning model is proposed in [108]. This model consists of three parallel layers: a segmentation branch that returns the lesion mask, a binary classification branch for melanoma detection, and a binary classification branch for seborrhoeic keratosis detection. The input of the network are the images to which several labels describing different lesion characteristics are associated, whereas the output provides the binary mask of the lesion, the probability of belonging to the melanoma class, and the probability of belonging to the seborrhoeic keratosis class. The model is implemented based on the GoogLeNet architecture, which is common to all three branches; the U-Net is used for segmentation and two FC layers are added for the classification branches.

In [109], a new DL architecture called NABLA-N Network for lesion segmentation, and the inception recurrent residual convolutional neural network (IRRCNN) model for skin cancer lesion classification are proposed. The classification network consists of three recurrent residual units followed by subsampling layers. At the end of the model, a GAP layer is used, which helps to significantly reduce the number of network parameters compared to a FC layer, followed by a softmax layer. The model is evaluated with and without DA, showing that performance increases significantly in the latter case.

In [110], multiple TL models based on XceptionNet, DenseNet201, ResNet50, and MobileNetV2 are tested. After training with the preprocessed and augmented images, the best model in terms of accuracy, precision, recall, and F1 score is the one based on ResNet50.

A combination of a multilabel deep feature extractor (ResNet50 backbone) with a clinically constrained classification chain to formulate the seven-point checklist algorithm based on the major and minor criteria and their respective weightings used by dermatologists is proposed in [111]. Each input, consisting of a clinical and a dermoscopic image, is associated with a label for the diagnosis (melanoma/non-melanoma) and seven labels for the evaluation criteria scores. Image features are extracted from the network, reduced in dimensionality by PCA, concatenated, and sent to the grading chain to obtain predictions on all seven-point checklists. The final score is the sum of all predictions weighted by the respective clinical weights (weight = 2 for major criteria and weight = 1 for minor criteria). A score greater than or equal to 3 produces a diagnosis of melanoma. By keeping

the criteria of the seven-point analysis, the proposed system could be more accepted by dermatologists as a human-interpretable CAD tool for automated melanoma detection.

In [112], several features are extracted from the skin lesions and a subsequent feed-forward neural network is used to perform classification by using the Levenberg Marquardt generalisation method (LM) to minimise mean square error. The extracted features are mean, standard deviation and skewness; entropy, mean and energy using the discrete 2D wavelet transform; contrast, similarity, energy and homogeneity using the GLCM.

A mixed skin lesion picture generated method based on Mask R-CNN (MSLP-MR) is implemented in [113] to augment the class of melanomas and reduce data imbalance. The augmented dataset is used to train models such as InceptionV4, ResNet, and DenseNet121, of which the latter is the best. Based on this observation, the DenseNet network is deepened by creating the DenseNet architecture145.

In [114], the optimal deep neural network driven computer-aided diagnosis for skin cancer detection and classification (ODNNsingle bondCADSCC) model is designed, which applies preprocessing based on Wiener filtering (WF), performs lesion segmentation with U-Net and extracts features with SqueezeNet. Finally, the improved whale optimization algorithm (IWOA) selects the parameters of the feed-forward DNN (FFNN) with three hidden layers that will be used for the effective detection and classification of skin cancer.

For the classification of melanoma, the TL on the SqueezeNet is used in [115], the optimal parameters of which are identified by using the bald eagle search (BES) method. In addition, a random oversampling method (ROS) followed by data augmentation is used to eliminate data imbalance. This approach, in addition to yielding excellent results in terms of accuracy, sensitivity, specificity, F1 score, and AUC, requires less training time than other pretrained networks including VGG19, GoogleNet, and ResNet50.

In [116], the authors propose a study on the effect of image size for skin lesion classification based on pretrained CNNs and transfer learning. After examining the classification performance of three well-established CNNs, namely EfficientNetB0, EfficientNetB1, and SeReNeXt-50, it is shown that image cropping is a better strategy than scaling and provides superior classification performance at all image scales from $224 \times 224$ to $450 \times 450$. Furthermore, for the classification of skin lesions the authors of the paper propose and evaluate a unique multiscale multi-CNN (MSM-CNN) fusion approach, which consists of assembling the results of three different fine-tuned networks, trained with cropped images at six different scales. After each of the three models (EfficientNetB0, EfficientNetB1 and SeReNeXt-50) has performed a prediction on the cropped images in six different formats, the average of the six classifications for each network is obtained and then the three final results are averaged again to obtain the final classification.

In [117], the authors propose a multiclass multilevel classification algorithm (MCML) for multiclass (healthy, benign, malignant, and eczema) classification of skin lesions and evaluate the use of traditional machine learning and an advanced deep learning aproach. In the first approach, after the steps of preprocessing, segmentation and feature extraction, an ANN with three hidden layers is used to perform classification. In the second approach, the TL is used, and a pretrained AlexNet model is modified, fine-tuned and retrained on the dermatology dataset. The best results are obtained with the DL approach.

In [118], a new deep-learning methodology is proposed to implement effective skin disease classification. After preprocessing and image segmentation, deep features are extracted by using Resnet50, VGG16, and Deeplabv3 and then concatenated. These concatenated features are transformed by using hybrid squirrel butterfly search optimization (HSBSO) and then passed to modified long short-term memory (MLSTM), where architecture optimization is performed by HSBSO itself to produce the final classified output.

This paper [119] proposes a self-supervised topology clustering network (STCN) by a transformation-invariant network with self-supervised maximum modularity clustering algorithm following topology analysis principle. A pre-trained ResNet50 is used as a feature-extraction module, and the image decoder in cycle GAN is used as a self-expression module. Finally, the feature vectors of the images are used to train a deep topology

clustering algorithm that performs clustering, and a softmax layer is added downstream of the feature vector to make the entire network capable of performing classification.

The authors in [120] propose a deep convolutional neural network (DCNN) model to perform accurate classification of skin lesions into malignant and benign. The CNN is pretrained on a large image dataset (ImageNet), and then fine tuned to a new dermatological dataset. In testing, the proposed model achieves good performance in terms of accuracy, precision, recall, and F1 score. This model is found to be more accurate when the pathology is in an early stage.

In [121], the authors present a framework for skin cancer classification that combines image preprocessing with a hybrid-CNN. The proposed CNN consists of three feature extraction blocks. The feature maps output from these blocks are sent to an FC layer either individually or concatenated with each other. Finally, the results are merged to provide the overall output.

In [122], a novel deep learning framework for segmentation and classification of skin lesions is proposed. In the classification phase, a 24-layer convolutional neural network architecture is designed, the best features of which are provided to softmax classifiers for final classification.

In [123], an average ensemble learning-based model is proposed to use five pretrained deep neural network models (ResNeXt, SeResNeXt, ResNet, Xception, and DenseNet) as the basis of the ensemble to classify seven types of skin lesions. The grid search method is used to find the best combination of the basic models and perform a weighted average combination, but it is shown that the models all behave more or less the same, except for DenseNet, and therefore the unweighted average combination can be used.

A novel deep convolutional neural network for the melanoma and seborrheic keratosis detection task is presented in [124]. The novelty of this approach is to use a pretrained ResNet18 network for classification of the original images, and four other pretrained AlexNet networks for classification of four new images obtained by applying Gabor wavelet filters with coefficients $0°$, $45°$, $67.5°$, and $112.5°$. The final decisions of the five classification networks are finally merged to improve the overall performance.

The authors of [125] propose a deep convolutional neural network, named Classification of Skin Lesion Network (CSLNet), to perform multi-class classification of skin lesions. The network consists of concatenated basic blocks with a total of 68 convolutional layers, each preceded by a batch normalization layer and a LeakyRelu layer. Finally, a Global Average Pooling layer precedes the last FC layer before the output layer.

In the paper [126] a deep convolutional ensemble neural network is created to perform classification of dermoscopic images into three classes: melanoma, nevus, and seborrheic keratosis. The classification layers of four different deep neural networks are fused, two pre-trained (ResNet and GoogLeNet), and two with weights initialized to random values (VGGNet and AlexNet). The final classification is obtained by performing the weighted sum of the maximal probabilities (SMP) of each network.

To classify melanoma images into malignant and benign, in [127] a pretrained MobileNetV2 network is used as the basis of the model and adds a global average pooling followed by two fully connected final layers. Evaluation of the model on four different datasets shows poor accuracy in classifying malignant lesions, a result likely related to the imbalance between classes. As it is designed, the proposed model can also be implemented on mobile devices.

In [128], the addition of features in the layers of a CNN is proposed. Specifically, features are extracted from segmented dermoscopic images and used as additional input to the CNN network layer. The handcrafted features, which include shape, color, and texture features (extracted by GLCM and scatter wavelet transform), and the features extracted by CNN are concatenated at the fully connected layer leading to high performance in classifying various skin lesions.

In [129], a deep convolutional neural network framework for multiclass classification of skin lesions is proposed, including the outcome of binary classification (healthy/diseased)

in the final probabilities. To accomplish this, the pretrained GoogLeNet-InceptionV3 network is used to perform multiclass and binary classification simultaneously, and the respective softmax outputs are merged on a support training layer. This layer multiplies the confidence of multiclass classification with the corresponding confidence of binary classification.

Table 5 summarizes the deep-learning methods previously described.

**Table 5.** Overview of cited works using DL approaches (results have been rounded). $F_{EX}$ and $F_{SE}$ abbreviations are used for feature extraction and selection, respectively. The symbol "-" indicates that no information is provided on a particular operation.

| Author & Year | Classification Task | Dataset | Data Augmentation | Methods Used | Cross Validation | Results |
|---|---|---|---|---|---|---|
| **Raza et al. [85]** 2022 | Binary: benign vs. malignant | ISIC archive | - | Parameter transfer of a pre-trained network to a CNN | - | ACC = 0.96 |
| **Guergueb et al. [86]** 2022 | Binary: benign vs. malignant | ISIC archive, ISIC2020 | Mixup and CutMix techniques | Ensemble of three pre-trained CNNs: EfficientNetB8, SEResNeXt10 and DenseNet264 | 3-fold | ACC = 0.989, SE = 0.962, SP = 0.988, AUC = 0.99 |
| **Shahsavari et al. [87]** 2022 | Multi-class: BCC, MM, N, SK | ISIC archive, PH$^2$ | Image rotation: 45, 90, 135, 180, 210; horizontal and vertical flipping | Ensemble of four pre-trained CNNs: GoogLeNet, VGGNet, ResNet and ResNeXt | - | ACC = 0.879 on ISIC, ACC = 0.94 on PH$^2$ |
| **Wu et al. [88]** 2022 | Multi-class: N/AN, MM, SK, BCC, DF, AK, VL | HAM10000 | Random clipping, flipping and ranslation | Use of TL on InceptionV3, ResNet50 and Denset201 | - | ACC train = 0.99, ACC val = 0.869 |
| **Thapar et al. [89]** 2022 | Binary: MM vs. non-MM | ISIC2017, ISIC2018, PH$^2$ | - | $F_{EX}$ and $F_{SE}$ (based on GOA) + custom CNN | - | ISIC2017: ACC 0.98= , SE = 0.96, SP = 0.99, PR = 0.97, F1 = 0.97; ISIC2018: ACC = 0.98, SE = 0.97, SP = 0.99, PR = 0.98, F1 = 0.97; PH$^2$: ACC = 0.98, SE = 0.96, SP = 0.99, PR = 0.97, F1 = 0.96 |
| **Kumar et al. [90]** 2022 | Binary: benign vs. malignant | ISIC archive | Resizing, vertical and horizontal flipping and rotation (45 degrees) | Pre-trained SqueezeNet re-trained by AWO algorithm | 5 and 9-fold | ACC = 0.925, SE = 0.921, SP = 0.917 |
| **Vanka et al. [91]** 2022 | Binary: benign vs. malignant | ISIC archive | - | Custom CNN | - | TPR = 0.94, TNR = 0.98, F1 = 0.96 |
| **Girdhar et al. [92]** 2022 | Multi-class: N/AN, MM, SK, BCC, DF, AK, VL | HAM10000 | Details are missing | Custom CNN | - | ACC = 0.963, REC = 0.96, F1 = 0.957 |
| **Montaha et al. [93]** 2022 | Binary: benign vs. malignant | ISIC archive | Brightness and contrast alteration of images | Custom shallow CNN | 5 and 10-fold | ACC = 0.987, PR = 0.989 |
| **Patil et al. [94]** 2022 | Multi-class: N/AN, MM, SK, BCC, DF, AK, VL | HAM10000 | - | Pre-trained DL method | - | ACC = 0.997 |
| **Tabrizchi et al. [95]** 2022 | Binary: MM vs. benign | ISIC2020 | Image rotation: 90, 180, 270 degrees; center cropping, brightness change, and mirroring | New DL model based on VGG16 | Leave-one-out | ACC = 0.87, SE = 0.852, F1 = 0.922, AUC = 0.923 |
| **Diwan et al. [96]** 2022 | Multi-class: N/AN, MM, SK, BCC, DF, AK, VL | HAM10000 | - | Custom CNN based on AlexNet | - | ACC = 0.878, SP = 962, PR = 0.787, REC = 0.774, F1 = 0.778 |

**Table 5.** *Cont.*

| Author & Year | Classification Task | Dataset | Data Augmentation | Methods Used | Cross Validation | Results |
|---|---|---|---|---|---|---|
| **Sharma et al. [97]** 2022 | Binary: benign vs. malignant | HAM10000 | - | Use of some pre-trained networks: VGG16, VGG19, DenseNet101 and ResNet101 | - | ACC = 0.848 |
| **Jojoa Acosta et al. [98]** 2021 | Binary: bening vs. malignant | ISIC2017 | Image rotation: 180 degrees; vertical flipping | Use of pre-trained ResNet52 in 5 different situations | - | ACC = 0.904, SE = 0.82, SP = 0.925 |
| **Romero Lopez et al. [99]** 2017 | Binary: benign vs. malignant | ISIC2016 | - | Use of VGG16 in 3 different situations | - | ACC = 0.813, SE = 0.787, PR = 0.797 |
| **Wei et al. [100]** 2020 | Binary: benign vs. malignant | ISIC2016 | Image rotation: 90, 180, 270 degrees; mirroring, center cropping, brightness change and random occlusion operations | Custom architecture based on MobileNet and DenseNet | - | MobileNEt ACC = 0.865, AUC = 0.832; DenseNEt: ACC = 0.855, AUC = 0.845 |
| **Safdar et al. [101]** 2021 | Binary: MM vs. benign | PH$^2$, MedNode, ISIC2020 | Affine Image Transformation and color Transformation approaches | Use of pre-trained ResNet50 and InceptionV3 | - | ACC = 0.934, SP = 0.965, PR = 0.895, AUC = 0.988 |
| **Ozturk et al. [102]** 2022 | Binary: benign vs. malignant | HAM10000, ISIC2019, ISIC2020 | - | Deep clustering approach. Custom CNNs based on VGG16, ResNet50, DenseNet169 and EfficientNetB3 | - | ACC = 0.98, SP = 0.999, PR = 0.961, REC = 0.98, F1 = 0.97, AUC = 0.709 |
| **Garcia [103]** 2022 | Multi-class: MM, benign, malignant | ISIC2019, PH$^2$, 7-point checklist dataset | - | Use of a meta-learning method and pre-trained ResNet50 | 3-fold | F1 = 0.53, Jaccard similarity index= 0.472 |
| **Nadipineni [104]** 2020 | Multi-class: MM, N, BCC, AK, SK, DF, VL, SCC | ISIC2019, 7-point checklist dataset | Random brightness, contrast changes, random flipping, rotation, scaling, and shear, and CutOut | Use of pre-trained MobileNet | 10-fold | ACC = 0.886 |
| **Chaturvedi et al. [105]** 2020 | Multi-class: N/AN, MM, SK, BCC, DF, AK, VL | HAM10000 | Image rotation, zoom, horizontal/ vertical flipping | Use of three pre-trained networks (EfficientNet, SENet and ResNet) in three different situations | - | ACC = 0.831, PR = 0.89, REC = 0.83, F1 = 0.83 |
| **Milton [106]** 2019 | Multi-class: N/AN, MM, SK, BCC, DF, AK, VL | HAM10000 | Image rotation, flipping, random cropping, adjust brightness and contrast, pixel jitter, Aspect Ratio, random shear, zoom, and vertical/horizontal shift and flip | Use of pre-trained networks: PNASNet-5-Large, InceptionResNetV2, SENet154 327 and InceptionV4 | - | ACC = 0.76 |
| **Majtner et al. [107]** 2018 | Multi-class: N/AN, MM, SK, BCC, DF, AK, VL | ISIC2018 | Image rotation, horizontal flipping | Combination of VGG16 and GoogLeNet pre-trained networks | - | ACC VGG16 = 0.801, ACC GoogLeNet = 0.799, ACC ensemble = 0.815 |
| **Yang et al. [108]** 2017 | Multi-class: MM vs. N and KS; MM and N vs. SK | ISIC2017 | - | Custom CNN based on GoogLeNet | - | AUC = 0.926, Jaccard index = 0.724 |
| **Alom et al. [109]** 2019 | Multi-class: N/AN, MM, SK, BCC, DF, AK, VL | HAM10000 | Horizontal/ vertical flipping | Custom CNN | - | ACC = 0.871 |

**Table 5.** *Cont.*

| Author & Year | Classification Task | Dataset | Data Augmentation | Methods Used | Cross Validation | Results |
|---|---|---|---|---|---|---|
| **Agarwal et al. [110]** 2022 | Binary: benign vs. malignant | ISIC archive | Re-scaling, shearing, vertical/horizontal flipping, zoom | Use of TL on XceptionNet, DenseNet201, ResNet50 and MobileNetV2 | - | ACC = 0.866, PR = 0.865, REC = 0.86, F1 = 0.862 |
| **Wang et al. [111]** 2021 | Binary: benign vs. malignant | 7-point checklist dataset | - | Custom CNN based on ResNet50 | - | ACC = 0.813, SE = 0.529, SP = 0.891 |
| **Choudhary et al. [112]** 2022 | Binary: benign vs. malignant | ISIC2017 | Based on Mask R-CNN | $F_{EX}$ + FFNN | - | ACC = 0.826, SE = 0.857, SP = 0.764, REC = 0.893, F1 = 0.824 |
| **CaoaJeng et al. [113]** 2021 | Binary: MM vs. benign | ISIC2017, ISIC2018 | - | Use of pre-trained models: InceptionV4, ResNet and DenseNet121 | 5-fold | ACC = 0.906, SE = 0.78, SP = 0.934, AUC = 0.95 |
| **Malibari et al. [114]** 2022 | Multi-class: N, MM, SK, BCC, DF, AK, VL, SCC | ISIC2019 | - | Custom DNN | - | ACC = 0.956, SP = 0.963, PR = 0.847, REC = 0.925, F1 = 0.884 |
| **Sayeda et al. [115]** 2021 | Binary: MM vs. benign | ISIC2020 | Random translation, scale, rotation, reflection, and shear | Use of pre-trained SqueezeNet | - | ACC = 0.98, SE = 1, SP = 0.97, F1 = 0.98, AUC = 0.99 |
| **Mahbod et al. [116]** 2020 | Multi-class: N/AN, MM, SK, BCC, DF, AK, VL | ISIC2016, ISIC2017, ISIC2018 | - | Assembling of pre-trained EfficientNetB0, EfficientNetB1 and SeReNeXt-50 | - | ACC = 0.96, PR = 913, AUC = 0.981 |
| **Hameeda et al. [117]** 2020 | Multi-class Single-level; multi-class Multi-level | ISIC2016, PH$^2$, DermIS, DermQuest, DermNZ | - | $F_{EX}$ + ANN and pre-trained AlexNet | - | ML: ACC = 0.64; DL: ACC = 0.96 |
| **Elashiri et al. [118]** 2022 | Multi-class classification | PH$^2$, HAM10000 | - | $F_{EX}$ using Resnet50, VGG16 and Deeplabv3 + modified LSTM | - | PH$^2$: ACC = 0.94, SE = 0.94, SP = 0.93, PR = 0.90, F1 = 0.92; HAM: ACC = 0.94, SE = 0.94, SP = 0.94, PR = 0.34, F1 = 0.5 |
| **Wang et al. [119]** 2018 | Multi-class: N/AN, MM, SK, BCC, DF, AK, VL | ISIC2018 | Image rotation, flipping, scaling, tailoring, translation, adding noise, and changing contrast | $F_{EX}$ using pre-trained ResNet50 and decoder in Cycle GAN + STCN | - | ACC = 0.79, AUC = 0.81 |
| **Ali et al. [120]** 2021 | Binary: benign vs. malignant | HAM10000 | Image rotation, random cropping, mirroring, and color-shifting using principle component analysis | Custom DCNN | - | ACC = 0.91, PR = 0.97, REC = 0.94, F1 = 0.95 |
| **Hasan et al. [121]** 2022 | Binary: MM vs. N; Multi-class: MM, N, SK and N/AN, MM, SK, BCC, DF, AK, VL | ISIC2016, ISIC2017, ISIC2018 | Image rotation (180, 270 degrees); gamma, logarithmic, and sigmoid corrections, and stretching, and shrinking of the intensity levels | Custom CNN | 5-fold | ISIC2016: AUC = 0.96, REC = 0.92, PR = 0.92; ISIC2017: AUC = 0.95, REC = 0.86, PR = 0.86; ISIC2018: AUC = 0.97, REC = 0.86, PR = 0.85 |
| **Khan et al. [122]** 2021 | Multi-class: N/AN, MM, SK, BCC, DF, AK, VL | HAM10000 | - | Custom CNN | - | ACC = 0.87, SE = 0.86, PR = 0.87, F1 = 0.86 |

**Table 5.** *Cont.*

| Author & Year | Classification Task | Dataset | Data Augmentation | Methods Used | Cross Validation | Results |
|---|---|---|---|---|---|---|
| **Rahman et al. [123]** 2019 | Multi-class: N/AN, MM, SK, BCC, DF, AK, VL | ISIC2019, HAM10000 | Image rotation (0–30 degrees), flipping, shearing (0.1), and zooming (90% to 110%). | Ensemble of 5 pre-trained models (ResNeXt, SeResNeXt, ResNet, Xception and DenseNet) | - | ACC = 0.87, PR = 0.87, REC = 0.93, F1 = 0.89, MCC = 0.87 |
| **Sertea et al. [124]** 2019 | Binary: MM vs. SK | ISIC2017 | Image rotation: 18, 45 degrees | Use of pre-trained ResNet18 and AlexNet | - | MM: ACC = 0.83, SE = 0.13, SP = 1, AUC = 0.96; SK: ACC = 0.82, SE = 0.17, SP = 0.98, AUC = 0.66 |
| **Iqbal et al. [125]** 2021 | Multi-class: N/AN, MM, SK, BCC, DF, AK, VL | ISIC2017, ISIC2018, ISIC2019 | Image rotation (30 to 30 degrees), translation (12.5% shift to the left, the right, up, and down), and horizontal/vertical flipping | Custom CNN | - | ISIC2017: ACC = 0.93, SE = 0.93, SP = 0.91, PR = 0.94, F1 = 0.93, AUC = 0.96; ISIC2018: ACC = 0.89, SE = 0.89, SP = 0.96, PR = 0.90, F1 = 0.89, AUC = 0.99; ISIC2019: ACC = 0.90, SE = 0.90, SP = 0.98, PR = 0.91, F1 = 0.90, AUC = 0.99 |
| **Harangi [126]** 2018 | Multi-class: MM, N, SK | ISIC2017 | Cropping of random samples from the images; horizontal flipping and rotation (90, 180, 270 degrees) | Use of two pre-trained networks (ResNet and GoogLeNet), and two networks with randomly initialized weights (VGGNet and AlexNet) | - | ACC = 0.87, SE = 0.56, SP = 0.79, AUC = 0.89 |
| **Indraswari et al. [127]** 2022 | Binary: benign vs. malignant | ISIC archive, ISIC2016, MedNode, PH² | - | Use of modify pre-trained MobileNetV2 | - | ISIC archive: ACC = 0.85, SE = 0.85, SP = 0.85, PR = 0.83; ISIC2016: ACC = 0.83, SE = 0.36, SP = 0.95, PR = 0.64; MedNode: ACC = 0.75, SE = 0.76, SP = 0.73, PR = 0.67; PH²: ACC = 0.72, SE = 0.33, SP = 0.92, PR = 0.67 |
| **Kotra et al. [128]** 2021 | Binary: MM vs. n; SK vs. SCC; MM vs. SK; MM vs. BCC; N vs. BCC | ISIC2016 | - | Injection of hand-extracted features into the FC layer of a CNN | - | MM vs. N: ACC = 0.93; SK vs. SCC: ACC = 0.95; MM vs. SK: ACC = 0.98; MM vs. BCC: ACC = 0.99; N vs. BCC: ACC = 0.99 |
| **Harangi et al. [129]** 2020 | Binary: healthy vs. diseased; multi-class: N/AN, MM, SK, BCC, DF, AK, VL | HAM10000 | Cropping of random samples from the images; horizontal/vertical flipping, rotation (90, 180, 270 degrees) and application of random brighten and contrast factors | Use of modify pre-trained GoogLeNet-InceptionV3 network | - | MM: ACC = 0.91, SE = 0.45, SP = 0.97, PR = 0.68, AUC = 0.81 |

### 4.3. ML/DL Hybrid Techniques

For the task of identifying and classifying skin cancer in dermoscopic images, in [130] a hybrid-dense algorithm is proposed. This algorithm consists of the extraction of skin lesion features with the pre-trained DenseNet121 network and the subsequent dimensionality reduction of the obtained vectors. Finally, classification is performed with the XGBoost classifier. The developed algorithm shows robustness in testing, so it is designed as a viable alternative in the identification of cancer-like diseases in skin lesions.

A mix of images, hand-extracted features, and metadata is used in [131] to perform a multiclass classification based on ensemble networks. Multiple multi-input single-output (MISO) models, obtained by replacing the backbones with EfficientNet networks B4 to B7, are trained with the images to extract features, whereas the hand-extracted features and metadata are used for training an MPL with two dense layers. The outputs of the networks are then sent to an ANN, consisting of two dense layers, which will perform the final classification.

In [132], 200 geometric features are extracted from the images, which are then injected into the last convolutional layer of two pretrained DL architectures (ResNet50 and DenseNet201). Both models are then used as feature extractors, sent to an SVM model for final prediction.

In [133], the efficiency of 17 pretrained CNNs used as feature extractors and 24 classifiers is examined. The best combinations are obtained by using DenseNet201 in combination with FineKNN or CubicSVM.

The combination of hand-coded features, sparse coding methods, and SVM with recent ML techniques (deep residual network and fully CNN) is presented in [134]. The features, extracted by hand, by sparse coding methods and by neural networks, are finally sent to an SVM classifier.

For skin lesion classification, the integration of handcrafted (HC) features (of texture, color, and shape) and features extracted by DL (using ResNet50V2 and EfficientNetB0) is also used in [135]. After obtaining vectors of the features (HC, from ResNet and Efficient-Net), they train the ANN itself on these three single vectors, on the combination of the HC vector and ResNet and on the combination of HC and EfficientNet, obtaining the best results with the latter combination.

In [136], the InceptionV3 network is used as a feature extractor (1000-dimensional vector obtained downstream of the penultimate layer of the network), and two different feed-forward neural networks (with two layers each and softmax activation function) for the classification of skin lesions into benign/malignant and melanocytic/non-melanocytic.

The problem of limited and unbalanced data is addressed in [137], in which the authors propose an approach that improves the model's ability to handle these problems. The classifications of the six models are then merged with the metadata associated with the images in the dataset and sent to an SVM classifier. In this paper, the authors show that ensemble learning significantly improves classification accuracy even from low accuracies for individual models, and that TL and the use of metadata have only a minor effect on the result obtained.

In [138], eight pretrained CNN models are used simultaneously to extract deep features from the images, and 10 different classifiers to perform the classification. The different couplings show that the DenseNet121 network with subsequent MPL achieves the highest performance in terms of accuracy.

In [139], an ensemble method that combines several DL feature extractors of skin lesions with an SVM classifier with RBF kernel is proposed. Feature extraction is performed by using the pretrained AlexNet, VGG15, ResNet18, and ResNet101 models, replacing the last layer with an FC to perform a binary classification (MM, SK). The feature vectors are then classified with an SVM model whose scores were subsequently mapped into probabilities by using logistic regression. The fusion of the prediction probability vectors of the different models leads to excellent results.

In [140], a novel midlevel feature learning method for skin lesion classification is proposed to use the pretrained ResNet50 and DenseNet201 models as feature extractors from the previously segmented dermoscopic images, perform dimensionality reduction of the feature vectors by PCA, and obtain the midlevel feature representation of these vectors. Finally, the midlevel features, obtained by learning the similarities between each sample and a set of reference images, are passed to an SVM classifier with a kernel radial basis function (RBF).

The authors of [141] propose a framework for automatic skin lesion recognition by using an aggregation of multiple pretrained convolutional networks (VGG-M + VGG16 + ResNet50). They call cross-net the network ensemble strategy to distinguish it from the traditional ensemble networks method. The output activation maps of each network are extracted as indicator maps to select local deep convolutional descriptors in the dermoscopic images and then the selected descriptors are concatenated into an information map and encoded by using Fisher vector (FV). This method encodes the aggregated descriptors into a global image representation to obtain more discriminating information than conventional methods. Finally, for identification of melanocytic lesion, a linear SVM classifier is applied. They perform two binary classifications: distinction melanoma vs. other diseases, and distinction seborrheic keratosis vs. other diseases.

Table 6 summarizes the ML/DL hybrid techniques previously described.

**Table 6.** Overview of cited works using ML/DL hybrid approaches (results have been rounded). $F_{EX}$ and $F_{SE}$ abbreviations are used for feature extraction and selection, respectively. The symbol "-" indicates that no information is provided on a particular operation.

| Author & Year | Classification Task | Dataset | Data Augmentation | Methods Used | Cross Validation | Results |
|---|---|---|---|---|---|---|
| **Carvajal et al. [130]** 2022 | Binary: MM vs. carcinoma | HAM10000 | - | $F_{EX}$ using pre-trained DenseNet121 + XGBoost classifier | - | ACC = 0.91, SE = 0.93, PR = 0.91, F1 = 0.91 |
| **Sharafudeen et al. [131]** 2022 | Multi-class: N/AN, MM, SK, BCC, DF, AK, VL, SCC | ISIC2018, ICIS2019 | - | $F_{EX}$ with EfficientNet networks B4 to 486 B7 and hand-extracted features + ANN | - | ISIC2018: ACC = 0.91, SE = 0.98, ISIC2019: ACC = 0.86, SE = 0.98 |
| **Redha et al. [132]** 2021 | Multi-class: N/AN, MM, SK, BCC, DF, AK, VL | ISIC2018 | Random crops and rotation (0–180 degrees), vertical/horizontal flips, and shear (0–30 degrees) | $F_{EX}$ using pre-trained DL architectures (ResNet50 and DenseNet201) + SVM | - | ACC = 0.92, SE = 0.88, SP = 0.97, AUC = 0.98 |
| **Benyahia et al. [133]** 2017 | Multi-class: healthy, benign, malignant, eczema; multi-level | ISIC2019, PH$^2$ | - | $F_{EX}$ using pre-trained DenseNet201 + FineKNN or CubicSVM classifiers | - | ISIC: ACC = 0.92, PH$^2$: ACC = 99 |
| **Codella et al. [134]** 2017 | Binary: benign vs. malignant | ISIC2016 | - | $F_{EX}$ by hand, by sparse coding methods and by Deep Residual Network (DRN) + SVM | 3-fold | SP = 0.95, PR = 0.65, AUC = 0.84 |
| **Bansal et al. [135]** 2022 | Binary: MM vs. non-MM | HAM10000, PH$^2$ | Image rotation, vertical/horizontal flipping, zoom, increased brightness and contrast, and noise addition | $F_{EX}$: hand-crafted, from ResNet and 503 EfficientNet + ANN | - | HAM10000: ACC = 0.95, SE = 0.95, SP = 0.95, PR = 0.95, F1 = 0.95; PH$^2$: ACC = 0.98, SE = 0.98, SP = 0.98, PR = 0.96, F1 = 0.97; |
| **Mirunalini et al. [136]** 2017 | Binary: benign vs. malignant; MM vs. non-MM | ISIC2017 | - | $F_{EX}$ with InceptionV3 + FFNNs | - | 1°: ACC = 0.72; 2°: ACC = 0.71; average-AUC = 0.66 |

**Table 6.** *Cont.*

| Author & Year | Classification Task | Dataset | Data Augmentation | Methods Used | Cross Validation | Results |
|---|---|---|---|---|---|---|
| **Qureshi et al. [137]** 2021 | Binary: benign vs. malignant | ISIC archive, ISIC2020 | - | Ensemble of six CNN + SVM | | F1 = 0.23 ± 0.04, AUC-PR = 0.16 ± 0.04, AUC = 0.87 ± 0.02 |
| **Gajera et al. [138]** 2022 | Binary: MM vs. non-MM | ISIC2016, ISICI2017, HAM10000, PH$^2$ | | $F_{EX}$ using pre-trained DenseNet121 network + MPL | 5-fold | ISIC2016: ACC = 0.81; ISICI2017: ACC = 0.81; HAM10000: ACC = 0.81; PH$^2$: ACC = 0.98 |
| **Mahboda et al. [139]** 2019 | Binary: MM vs. SK | ISIC2016 | - | $F_{EX}$ using pre-trained AlexNet, VGG15, ResNet18 and ResNet101 models + RBF-SVM | - | MM:SE = 0.812, SP = 0.785, AUC = 0.873; SK : SE = 0.933, SP = 0.859, AUC = 0.955 |
| **Liu et al. [140]** 2020 | Binary: MM vs. non-MM; SK vs. non-SK | ISIC2017 | - | $F_{EX}$ using pre-trained ResNet50 and DenseNet201 models + RBF-SVM | - | ResNet: ACC = 0.87, AUC = 0.89; DenseNet: ACC = 0.87, AUC = 0.89 |
| **Yu et al. [141]** 2020 | Binary: MM vs. others; SK vs. others | ISIC2016, ISIC2017 | Image rotation, flipping, translation, and cropping; color-based data augmentation | $F_{EX}$ from CNNs + linear SVM | - | MM ISIC2016: ACC = 0.87, SE = 0.6, SP = 0.85, PR = 0.69, AUC = 0.86; MM ISIC2017: ACC = 0.84, SE = 0.61, SP = 0.90, PR = 0.63, AUC = 0.84; SK ISIC2017: ACC = 0.92, SE = 0.80, SP = 0.94, PR = 0.82, AUC = 0.95. |

## 5. Discussion and Conclusions

Skin cancer is one of the most common cancers in the world with a high mortality rate. Early identification and diagnosis of skin lesions is essential to determine the best treatment for the patient and to increase the survival rate in the case of cancerous lesions. Diagnosis of this disease is conducted manually by more or less experienced dermatologists, but it proves to be time consuming and difficult. By using CAD systems, this procedure can become much easier, faster, and more accurate.

This systematic literature review aims to provide an overview of the use of machine learning and deep learning in dermatology to help future researchers. Scientific publications published between 2012 and 2022 related to ML and DL approaches for the detection and classification of skin lesions were selected. The searches, conducted in the arXiv and Science Direct databases, resulted in the selection of 68 research articles that focused on skin lesion classification using images from public datasets and reported the results obtained in terms of model performance. Having chosen the use of public datasets among the inclusion criteria, there are no papers prior to 2016. Furthermore, more than half of the articles on ML were published from 2020 to 2022, and more than half of the articles on DL and ML/DL were published from 2021 to date. Overall, 70% of the papers selected for this article have been published in the past two years. The use of public datasets for model training and validation allows comparison of work, a key point of scientific research. An analysis of the datasets used in the papers cited in this review article shows that the HAM10000 dataset and the ISIC archive are the most frequently chosen datasets for training and testing skin lesion classification models (Figure 4). Of the latter, moreover, the 2016 and 2017 versions are the most frequently used over the past decade.

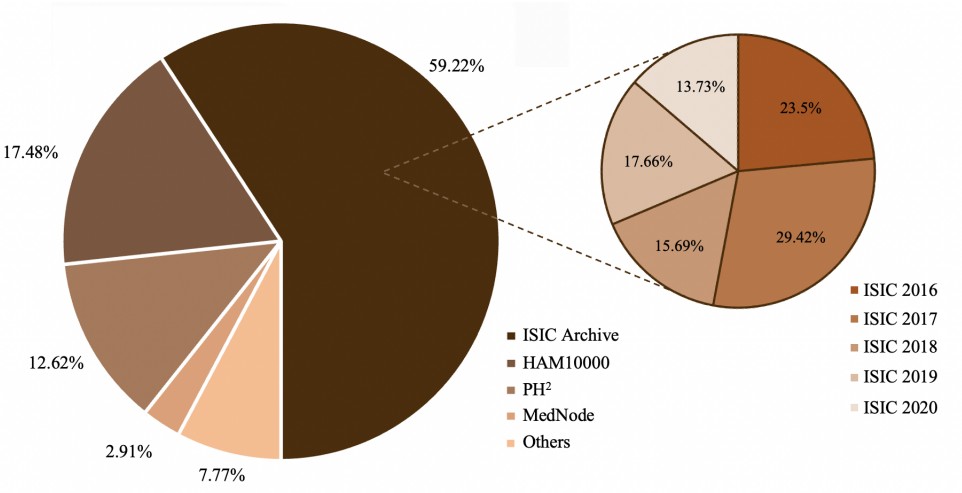

**Figure 4.** Analysis of datasets used. The "others" category includes DermIs, DermQuest, IDS, 7 point check list and DermNZ datasets.

The research conducted shows that the most widely used ML classifier is the SVM model (Figure 5), while pretrained convolutional neural networks account for the majority of DL and ML/DL approaches (Figure 6), and that among the many solutions identified, those based on DL represent the majority. Indeed, deep CNNs hold great promise for improving the accuracy of skin lesion identification and classification.

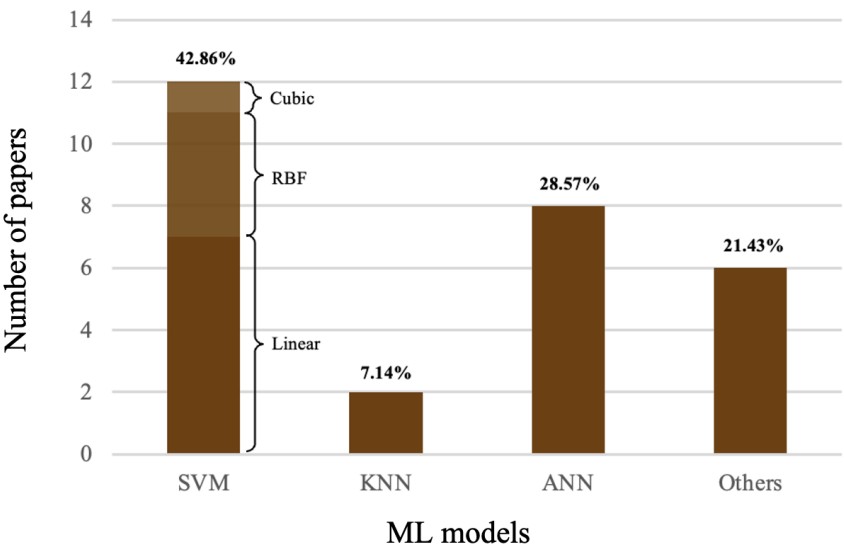

**Figure 5.** Analysis of the most used ML models. The "others" category includes the random forest, linear discriminant analysis, naive Bayes, RUSBoost, and XGBoost classifiers.

The results obtained, quantified through the metrics of accuracy, sensitivity, specificity, precision, recall, F1 score, and AUC, show that both ML and DL models developed in recent years—aimed at supporting diagnostic decisions and not replacing physicians—show high potential in skin lesion classification. It must be considered, however, that in the context of critical systems where errors are not allowed, such as in the medical field, there is an increasing demand for the comprehensibility of the algorithms used. This area of proposing models that can explain their own behavior is known as explainable AI (XAI) and has been the subject of numerous studies in recent years, including in the area of skin lesion classification [142–146]. In order for physicians to trust AI, the way in which machines

make decisions must be made clear. However, recent advances in the area of automated skin lesion classification bode well for the introduction of CAD systems into clinical practice in the not-too-distant future.

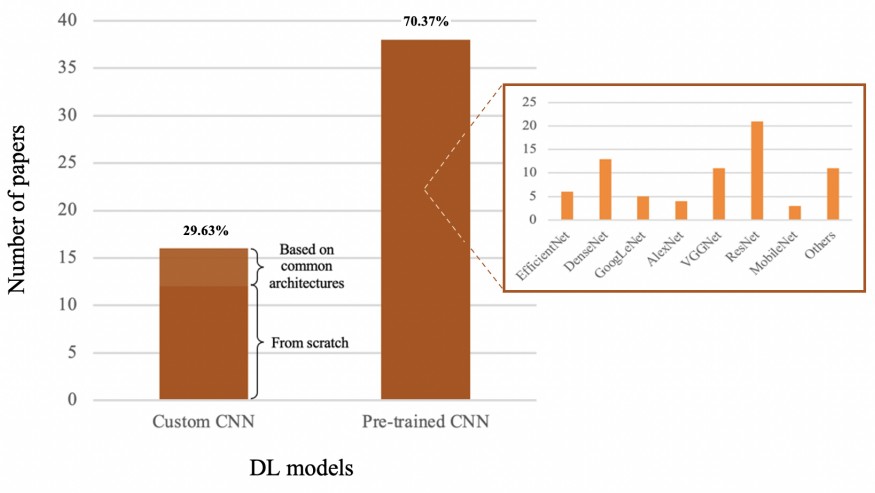

**Figure 6.** Analysis of the most used DL models, distinguishing between custom CNNs and pre-trained networks.

**Author Contributions:** Conceptualization, F.G., P.S., F.M., F.B., C.C. and M.T.; methodology, F.G.; literature search, F.G.; writing—original draft preparation, F.G.; writing—review and editing, P.S., F.M., F.B., C.C., M.T., F.M. and G.P.; supervision, F.F., L.P. and G.P. All authors have read and agreed to the published version of the manuscript.

**Funding:** This research received no external funding.

**Institutional Review Board Statement:** Not applicable.

**Informed Consent Statement:** Not applicable.

**Data Availability Statement:** Not applicable.

**Acknowledgments:** Not applicable.

**Conflicts of Interest:** The authors declare no conflict of interest.

## Abbreviations

The following abbreviations are used in this manuscript:

| | |
|---|---|
| SC | Skin Cancer |
| MM | Melanoma |
| BCC | Basal Cell Carcinoma |
| SCC | Squamous Cell Carcinoma |
| SK | Seborrheic Keratosis |
| CAD | Computer-Aided Diagnosis |
| ML | Machine Learning |
| DL | Deep Learning |
| ANN | Artificial Neural Network |
| CNN | Convolutional Neural Network |
| DA | Data Augmentation |
| TL | Transfer Learning |
| ACC | Accuracy |
| SE | Sensitivity |
| SP | Specificity |
| PR | Precision |
| REC | Recall |
| AUC | Area Under the ROC (Receiver Operating Characteristic) Curve |

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
