# Peer review of "Machine Learning Approaches for Skin Cancer Classification from Dermoscopic Images: A Systematic Review"

_algorithms, doi:10.3390/a15110438_

Round 1

Reviewer 1 Report

The authors prepared a survey paper and discuss Machine Learning Approaches for Skin Cancer Classification. The paper is well-organized and well-structured. My feedback and comments are as follows:

1) The authors should open a new Section and discuss and explain several machine learning methods (e.g., Random Forest, Support Vector Machines, ANN, etc.) and Convolutional Neural Networks (e.g. convolution layer, pool layer, etc. ). 

2) The authors should discuss the challenges, opportunities, and difficulties of skin cancer classification using ML and DL methods.

3) The authors should also discuss the application of the usage of explainable AI in skin cancer classification.

3) The authors can extend the related work section by including other research papers on different cancer diagnoses, such as breast cancer etc. 

Cancer Diagnosis With the Aid of Artificial Intelligence Modeling Tools

Survey on recent cancer classification systems for cancer diagnosis

MITNET: a novel dataset and a two-stage deep learning approach for mitosis recognition in whole slide images of breast cancer tissue

Breast Cancer Diagnosis Using Deep Learning Algorithm

Cancer Diagnosis Based on Combination of Artificial Neural Networks and Reinforcement Learning

Reviewer 2 Report

Dear authors,

congratulations on the interesting design and set of selected papers. It is an interesting collection of papers but I feel it still has a long way to go. Considering machine learning in general if you selected these papers based on inclusion/exclusion filters, I am still not aware which ones they were. Also, I am not convinced these papers have clearly and thoroughly described the procedures performed. Some of these papers seem to report only accuracy metrics which is not useful at all without additional information about class balance and train/test distribution.

Major concerns:

1) Figure 2) is general pipeline by which you can describe most of the image analysis tasks. Please include at least the general tools used, most used algorithms and techniques. For features add the most common and mostly repeated ones. For final classifications would be fine to add the most successful ones based on AUC or F1 score.

2) I like the focus on datasets, but I am completely missing information about which of the approaches are openly published and where I can find them. This is about 40% value of a review like this.

3)  In the overview tables I am completely missing information about the number of classes used, train/test distribution, cross-validation used, final classification method, and evaluation metrics used for training. Please include them or add information if it is missing in original paper. Without this, I am unable to decide if the technique is used properly and if it is useful.

4) In the table with deep learning please add what kind of augmentation was used for training.

5) I am completely missing a part with limitations of researched methods. Please add it. Even if it is general.

6) It will be quite useful if you add visual aids like charts of the most used databases, most used classifiers, and some graphical comparison of results and used approaches.

I believe after adding these points the paper will have good value and will be able to go through review with small edits.

At the moment it is, unfortunately, missing too many crucial parts. But I am looking forward to the improved version!

Reviewer 3 Report

Reviewer comments

It is essential to identify and diagnose the skin lesions in the early stage in order to determine the treatment and increase the survival rate of skin cancer. However, the manual diagnosis, which is time-consuming and difficult, relies on the experience and subjectivity of the physician. Therefore, it is desirable to develop computer aided diagnostic systems for the skin cancer diagnosis.

The manuscript is well-written and well-organized. The objective is well-articulated and reached. The figures and tables are presented in a clear and appropriate manner and are consistent with the description in the text. The results and analysis presented in the manuscript are interesting for this field and Algorithms is the appropriate journal to submit it. But there are still some points that the authors should consider, as described in the following. Also, some suggestions are provided, in case the authors consider them interesting to carry out.

In line 11-12, “only those documents were selected that clearly and completely described the procedures performed and reported the results obtained”. If the authors want to use a relative clause here, a proper way can be “only those documents that clearly and completely described the procedures performed and reported the results obtained were selected”.

In line 37, the abbreviation is cSCC. However, this abbreviation is SCC in Figure 1(b). Please keep them consistent.

In line 149, “The authors of [? ] use the lesion masks already provided with the images to extract 150 features (18 for shape, 72 for colour and 420 for texture) which are then manipulated to … ”. I guess the question mark at the beginning of this sentence should be the reference [50]. Please correct it. And there should be 510 features, not 150 features, because 18+72+420=510.

In this manuscript, the authors mostly use the present tense. However, the past tense is used at some places (e.g., line 158, 177, 192, 312, 331, 522, 525, 527). It is recommended to check the text and keep the tense consistent in the manuscript.

In Table 3, Oliveira et al. [? ]. The question mark should be the reference [50].

In line 225-226, “An Ensemble of Deep Model (SLDEP) is created using four different CNNs (GoogLeNet, ResNet and ResNeXt) to perform multi-class classification based on majority voting.” There are only three CNNs listed here, not four different CNNs. The missing one should be VGGNet.

In line 253, “In [? ], a new DL model is proposed based on …”. The question mark at the beginning of this sentence should be the reference [67].

In line 270, “new adversarial example images are created that maximise the loss for the input image”. A better way may be “new adversarial example images are created to maximise the loss for the input image”.

In line 289-290, “An architecture is created that takes as input a pair of images (malignant/benign) and uses a light network pre-trained on the ImageNet dataset …”. A better way may be “An architecture is created to take a pair of images as input and use a light network pre-trained on the ImageNet dataset …”.

In line 326-327, “In [? ], the MobileNet network pre-trained on images from the ImageNet dataset and optimised on dermatological images is used.” The question mark should be the reference [77].

In line 344-346, “a new DL architecture called NABLA-N Network for lesion segmentation is proposed, and the Inception Recurrent Residual Convolutional Neural Network (IRRCNN) model for skin cancer lesion classification.” Do the authors mean NABLA-N Network is proposed for lesion segmentation and IRRCNN is proposed for lesion classification? The second part of this sentence doesn’t have a verb.

In line 434-436, “an average ensemble learning-based model is proposed that uses 5 pre-trained deep neural network models …”. A better way should be “an average ensemble learning-based model is proposed to use …”.

In line 457-458, “a pre-trained MobileNetV2 network is used as the basis of the model and add a global average pooling …”. Here “add” should be “adds”.

In line 465, “, and and the features extracted by CNN …”. There are two “and” in this sentence.

In Table 4, there are question marks in Tabrizchi et al. [? ] and Chaturvedi et al. [? ].

 In line 500-501, “For the classification of skin lesions, also in [107] use the integration of hand crafted (HC) features (of texture, colour and shape) and extracted by DL …”. There is no subject in this sentence. And it should be “… features extracted by DL”, not “features and extracted by DL”.

In line 521, “an ensemble method is proposed that combines several DL feature extractors …”. A better way should be “an ensemble method is proposed to combine several DL feature extractors”, or “an ensemble method that combines several DL feature extractors … is proposed”.

In line 528-531, “a novel mid-level feature learning method for skin lesion classification is proposed that uses …, performs …, and obtains …”. We should say “… is proposed to use …, perform …, and obtain …”, or “the authors in [112] propose a method which uses …, performs …, and obtains …”.

In line 566-568, “The research conducted shows that among the most widely used ML classifiers is the SVM model, …”. We can say “the most widely used ML classifier is the SVM model”.

Round 2

Reviewer 1 Report

Based on my comments and feedback, the authors improved the manuscript extensively. It is ready for publication as is.